# Safety of Mechanically Fibrillated Cellulose Nanofibers (CNFs) by Inhalation Exposure Based on TG412

**DOI:** 10.3390/nano15030214

**Published:** 2025-01-28

**Authors:** Yoshihiro Yamashita, Akinori Tokunaga, Koji Aoki, Tamotsu Ishizuka, Satoshi Fujita, Shuichi Tanoue

**Affiliations:** 1Research Center for Fibers and Materials, University of Fukui, 3-9-1 Bunkyo, Fukui 910-8507, Japan; 2Life Science Research Laboratory, School of Medical Sciences, University of Fukui, 23-3, Matsuokashimoaizuki, Eiheiji-cho, Fukui 910-1193, Japan; 3Department of Pharmacology, Faculty of Medicine, University of Fukui, 23-3, Matsuokashimoaizuki, Eiheiji-cho, Fukui 910-1193, Japan; 4Department of Respiratory Medicine, Faculty of Medical Sciences, University of Fukui, 23-3, Matsuokashimoaizuki, Eiheiji-cho, Fukui 910-1193, Japan; 5Department of Frontier Fiber Technology and Science, Faculty of Engineering, University of Fukui, 3-9-1 Bunkyo, Fukui 910-8507, Japan

**Keywords:** OECD TG412, fib-CNFs, in-vivo, rat, inhalation exposure, NOAEL

## Abstract

An investigation into the acute toxicity of mechanically fibrillated cellulose nanofibers (fib-CNFs), with a fiber length ranging from 500 to 600 nm, was conducted in accordance with the OECD TG412 guidelines. In this study, rats were exposed to fib-CNFs via nasal inhalation for 6 h daily over a 28-day period. The highest exposure concentration was set at 35 mg/m^3^, with intermediate and low concentrations at 7.0 mg/m^3^ and 1.5 mg/m^3^, respectively. No significant differences were observed in body weight, hematological parameters, or biochemical profiles between the fib-CNF-exposed groups and the control group. However, the histopathological examination of lung tissue revealed elevated macrophage counts in both the alveolar spaces and lymph nodes, accompanied by a significant increase in lung weight. The most severe effects were observed in the high-concentration group, while the low-concentration group exhibited only mild inflammatory changes. Based on these findings, the no observable adverse effect level (NOAEL) for the acute toxicity of fib-CNFs is estimated to be below 1.5 mg/m^3^.

## 1. Introduction

Cellulose nanofibers (CNFs) are promising for diverse applications, including in thermoplastic composites, automotive components, and food additives [1]. However, it is essential to assess the occupational health risks associated with inhaling CNF aerosols or dry CNF powders. Inhalation is a primary exposure route in workplaces with elevated airborne CNF levels. It is well-established that inhaling particulates like asbestos, silica, or carbon dust can cause pulmonary irritation and elicit immune responses [2,3]. Although workers might consider CNFs safe due to their natural, plant-based origin, their nanoscale dimensions and fibrous morphology necessitate a thorough evaluation of their potential to induce respiratory inflammation and immune reactions upon inhalation.

In recent years, in vitro assays have been employed to assess the biocompatibility of cellulose nanofibers (CNFs), aiming to reduce reliance on animal models [4,5,6,7,8,9,10,11,12,13,14]. However, many researchers contend that in vivo studies remain indispensable for evaluating the pulmonary toxicity of inhaled CNFs. This perspective is supported by Harvard medical school [15], which recommends conducting in vivo studies when in vitro results indicate compromised epithelial barrier integrity, the release of pro-inflammatory mediators, or effects on immune cells or responses.

As a result, there has been an increase in reports of animal experiments on the respiratory safety of cellulose nanofibers (CNF), cellulose nanocrystals (CNC), and TEMPO-oxidized cellulose nanofibers (TEMPO-CNF) [16,17,18,19]. Catalán et al. [20] conducted an investigation into the pulmonary effects of a single pharyngeal aspiration of TEMPO-CNFs in murine models. Fibers measuring 500–1000 nm in length and 10–25 nm in diameter were administered at doses of 10, 40, 80, or 200 µg per mouse. Even after a single exposure, TEMPO-CNFs accumulated in the bronchi and alveoli, where macrophages internalized the fibers.

Park et al. [21] evaluated the impact of mechanically disassembled CNFs, termed fib-CNFs, which measured 142 ± 14 nm in length and 56 ± 14 nm in diameter. These fibers, also known as mechanolytic CNFs, mechanolyzed CNFs, or BiNFi-s ultra-short CNFs, were tested for their immunological and inflammatory effects in lung tissue 14 days post-exposure. Observations revealed significant inflammatory responses, including elevated counts of mononuclear phagocytes, polymorphonuclear leukocytes, and bronchoalveolar lavage (BAL) lymphocytes. Increased cytokine and chemokine expression, along with elevated lactate dehydrogenase (LDH) activity, indicated cellular stress and tissue inflammation.

Fujita et al. [22,23] investigated the pulmonary inflammatory responses in rats following exposure to phosphorylated cellulose nanofibers (CNFs), TEMPO-oxidized CNFs, mechanically fibrillated CNFs (mechanolyzed), and multi-walled carbon nanotubes (MWCNTs). Each material was administered via a single intratracheal instillation. Subsequent analyses included bronchoalveolar lavage fluid (BALF) assessment, histopathological examination, and comprehensive pulmonary gene expression profiling.

The phosphorylated and TEMPO-oxidized CNFs exhibited fiber diameters of 7–8 nm and lengths of 800–1000 nm, whereas the mechanolyzed CNFs had diameters of 21.2 nm and lengths of 1700 nm. The findings indicate that mechanolyzed CNFs induced a milder acute alveolar inflammatory response compared to phosphorylated and TEMPO-oxidized CNFs. Moreover, the pulmonary inflammatory response correlated with fiber dimensions; specifically, the shorter TEMPO-oxidized CNFs demonstrated greater toxicity than the longer mechanolyzed CNFs.

Tatrai et al. [24] reported that a single intratracheal administration of 15 mg of cellulose nanofibers (CNFs) in rats led to fibroplastic granulomatous bronchopneumonia and elevated immunoglobulin A (IgA) levels in bronchoalveolar lavage fluid (BALF). Over time, moderate fibroplastic alveolitis developed, accompanied by damage to type I pneumocytes and the incomplete repair of type II pneumocytes. This impaired repair process compromised the alveolar epithelium, initiating pulmonary fibrosis and ultimately resulting in the collapse of the alveolar-capillary functional unit.

Similarly, Čolić et al. [25] observed that, while nanocellulose is generally considered a low-toxicity, biocompatible material, cellulose nanocrystals (CNCs) specifically induced an inflammatory response following pulmonary exposure. Comparable inflammatory reactions were noted in vitro using macrophage cell lines. Notably, surface modification of CNCs with various chemical groups significantly reduced or redirected this inflammatory response toward anti-inflammatory pathways.

Many of the aforementioned toxicity studies employed single intratracheal administrations of cellulose nanofibers (CNFs). While this approach is straightforward, it may confound inflammation resulting from the test material with reactions to the substantial volume of aqueous CNFs solution introduced into the pulmonary system. Consequently, more rigorous and ecologically valid inhalation toxicity assessments are warranted, adhering to the Organization for Economic Co-operation and Development (OECD) Test Guideline (TG) 412 [26].

OECD TG 412 is designed to comprehensively evaluate the toxicity of chemicals via inhalation over a subacute period of 28 days, providing robust data for the quantitative inhalation risk assessments. Typically, the main study includes groups of five male and five female rodents exposed to the test chemical at three or more concentration levels for 6 h per day over 28 days (4 weeks), alongside a group exposed to filtered air (negative control) or solvent (solvent control). The animals are then sacrificed within 24 h after the exposure period ends, and bronchoalveolar lavage (BALF) is performed on all test animals.

Implementing TG 412 necessitates sophisticated laboratory infrastructure, as it requires controlled exposure systems to limit the test substance to the nasal region of animals for 28 days. Consequently, this guideline has predominantly been applied to chemicals suspected of significant toxicity, and has seldom been utilized in CNFs testing.

The nose-only exposure method involves restraining animals for up to 6 h daily, as specified in OECD Test Guideline TG 412 [27,28]. While this technique ensures precise dosing, it can induce stress in the subjects. Conversely, the whole-body exposure method is less stressful; however, maintaining consistent temperature and humidity within the exposure chamber is challenging due to the water-dispersed nature of cellulose nanofibers (CNFs). Additionally, the nose-only exposure system facilitates the continuous removal of exhaled air and the consistent introduction of fresh CNF aerosols into the exposure chamber throughout the testing period. Given these considerations, the nose-only inhalation toxicity test method, compliant with TG 412, was employed in this study. Notably, there have been no prior reports of CNF testing fully adhering to TG 412 guidelines.

The most commonly utilized cellulose nanofibers (CNFs) include TEMPO-oxidized CNFs and mechanically fibrillated CNFs. Given that TEMPO-oxidized CNFs undergo the chemical modification of mechanically processed CNFs, we opted to assess the inhalation safety of mechanically fibrillated CNFs. The physicochemical properties of CNFs, such as fiber length and diameter, vary based on the source material and processing techniques. However, limited research has been conducted on the correlation between fiber dimensions and the inhalation toxicity of mechanically processed CNFs. Notably, a study involving a single intratracheal administration of mechanically fibrillated CNFs (fib-CNF) in mice indicated transient weight loss and increased lung weights, suggesting adaptive responses rather than overt toxicity. This underscores the need for further investigations to elucidate the relationship between fiber morphology and pulmonary toxicity.

In this study, we focused on the mass median aerodynamic diameter (MMAD) of cellulose nanofibers (CNFs) suitable for inhalation by rats, which is crucial for nasal exposure testing. Longer mechanically processed CNF fibers are challenging to aerosolize, leading to lower aerosol concentrations. As noted by Fujita et al., fibers shorter than 1000 nm, such as TEMPO-oxidized CNFs, may exhibit increased toxicity. Therefore, we selected BiNFi-s ultra-short CNFs, with an average fiber length of 624 nm, for this study. Nasal exposure tests were conducted at the chosen concentrations using these mechanically processed CNFs.

## 2. Methods

### 2.1. Fib-CNFs Sample (Test Material)

Commercially available samples of mechanically processed fib-CNFs (“BiNFi-s” ultra-short FMa1002, with a 2 wt. % concentration in aqueous solution, lot number D1089401) were obtained from Sugino Machine Co., Ltd. (Tonami, Toyama, Japan). The average fiber length and diameter were determined using field emission scanning electron microscopy (FE-SEM, JSM-7600F, JEOL, Tokyo, Japan) and transmission electron microscopy (TEM, Hitachi H-7650, operating at 100 kV, Tokyo, Japan). Ultra-purified water (collected fresh on the day of use from a Milli-Q IQ 7005 system, Merck KGaA, Darmstadt, Germany) was used as the medium. The solution preparation procedure was as follows:(a)Weigh the required amount of test substance using an electronic balance (ME3002, Mettler Toledo Co., Ltd., Tokyo, Japan) with a minimum readability of 10 mg,(b)Add the medium to the weighed test substance and adjust the volume to the predetermined amount;(c)Prepare a 0.5% (wt/vol) aqueous solution of the test substance by gently inverting and mixing;(d)After preparation, homogenize the solution using a homogenizer (model MH-1000, As One Corporation, Osaka, Japan) at 9000 to 11,000 rpm for 20 min.

### 2.2. Test Animals

In this study, 26 male and 26 female rats (Crl(SD) SPF) were purchased from Jackson Laboratory Japan, Inc. Yokohama, Japan. At the time of purchase, the rats were 7 weeks old. Upon arrival, they underwent a 5-day quarantine period, during which no abnormalities were observed in their general condition. The rats were housed in an air-conditioned room with an air exchange rate of 6 to 20 times per hour. The temperature was maintained between 19.0 °C and 25.0 °C, with relative humidity ranging from 35.0% to 75.0%, and a 12 h light–dark cycle (from 7:00 a.m. to 7:00 p.m.). They were fed ad libitum with CR-LPF (Oriental Yeast Co., Ltd., Tokyo, Japan; radiation-sterilized; lot numbers 220713 and 220810). Feeding was restricted during the 6 h fib-CNFs exposure periods, during fresh urine collection, and from 5:00 p.m. on the day before the planned autopsy (planned range: 4:30 p.m. to 5:30 p.m.) until the autopsy was conducted. Tap water, filtered through a 5000 nm filter and irradiated with ultraviolet light, was provided ad libitum, except during the inhalation exposure period.

The acclimation period lasted from the animals’ arrival until the day before the exposure test began, at which point the rats were 8 weeks old. At the start of the exposure, the body weights of the male rats ranged from 292.1 to 345.4 g, while the female rats ranged from 213.3 to 258.8 g. It was confirmed that the body weights of all animals at the start of exposure were within ±20% of the mean body weight for both males and females.

Animals were randomly assigned to groups using weight-stratified random sampling to ensure that the mean body weight of each group was approximately equal. In total, 24 male and 24 female rats were used in the study. Animal identification was performed as follows.

Before grouping: A number (the last one or two digits of the animal identification number) was marked on the tail using an oil-based pen.

After grouping: A microchip was implanted subcutaneously in the dorsal region, and a microchip reader (IPT-300 and DAS-8007; Bio Medic Data Systems Inc., Seaford, DE, USA) was used for identification. Table 1 presents the group assignments for the rats.

### 2.3. Exposure Concentrations and Rationale

In a preliminary study, the maximum achievable concentration of mechanically processed fibrous cellulose nanofibers (fib-CNFs) in the inhalation apparatus was determined to be 7.44 mg/L when prepared as a 0.5% aqueous solution. To evaluate the subacute inhalation toxicity of the test substance, the exposure concentration was set at 7.0 mg/L, which is technically adjustable to the maximum achievable concentration.

The target exposure concentrations in the study were 0.3 mg/L, 1.4 mg/L, and 7.0 mg/L, using 0.5% aqueous solutions of the test substance. These values are more easily understood when expressed in cubic meters rather than liters. The solid fib-CNFs content in 1 cubic meter of air corresponds to mist weights of 0.3, 1.4, and 7 mg/L, which will hereafter be expressed as 1.5, 7.0, and 35 mg/m^3^ (fib-CNFs solid weight). The control group was exposed to a mist of purified water under the same conditions as the high-exposure concentration group.

#### 2.3.1. Nasal Exposure Apparatus

In this study, we utilized a nasal inhalation exposure system to assess the safety of the test substance for human inhalation. The apparatus used for administering the test substance is depicted in Figure 1. Following established protocols from similar studies, each rat was individually placed in a holding tube designed for nasal inhalation exposure (Muenster Ltd., Muenster, Germany). The exposure was conducted using a Flow-Past type inhalation chamber (Muenster Ltd.), featuring a stackable design with 16 exposure ports per tier. For this study, a single-stage configuration inhalation chamber was employed. The supply rate of the test atmosphere to the intake chamber was set at 20.0 L/min (with a planned range of 16.0 to 22.0 L/min). The exhaust flow rate was maintained at approximately 10% less than the supply rate to ensure that the animals were exposed to the test atmosphere while maintaining positive pressure in the inhalation chamber.

#### 2.3.2. Generation of the Test Substance

A container holding the test sample or purified water (obtained on the day of use from a Milli-Q^®^ IQ 7005 purification system) was connected to a two-fluid nebulizer (NB-2N; Sibata Scientific Technology Ltd., Saitama, Japan). A mist of the exposure sample or purified water was generated by supplying a predetermined flow rate of compressed air. This mist served as the test atmosphere and was continuously supplied to the inhalation chamber. The exhaust air from the inhalation chamber was filtered before being released into the atmosphere. 

#### 2.3.3. Exposure

The rats were exposed to test atmospheres containing purified water (control, 0 mg/m^3^), 1.5 mg/m^3^ fibrous cellulose nanofibrils (fib-CNFs), 7.0 mg/m^3^ fib-CNFs, or 35 mg/m^3^ fib-CNFs (Table 1). Exposure commenced by placing the rats in restraining tubes inside the inhalation chamber at least 5 min after initiating the test atmosphere (observed times ranged from 5 to 30 min). This waiting period allowed the test atmosphere to achieve concentration equilibrium.

Given the inhalation chamber’s volume (approximately 2.5 L) and the test atmosphere’s supply flow rate (16–20 L/min), the estimated concentration equilibrium time (t95) was approximately 20–30 s. To ensure stability in the concentration of the test substance, exposure was defined as starting no earlier than 5 min after the test atmosphere was initiated.

Following 6 h of exposure, the rats were removed from the inhalation chamber, and the session was concluded. The exposure procedure was conducted once daily for 6 h, 5 days per week, over a 28-day period. One day after the final exposure, the rats were euthanized via exsanguination from the abdominal aorta under deep isoflurane anesthesia.

#### 2.3.4. Determination of Exposure Concentration

The theoretical concentration (*C_n_*) was calculated based on the volume of the test substance’s aqueous solution introduced into the inhalation chamber and the chamber’s airflow rate. This approach aligns with standard practices for estimating inhalation exposure concentrations.$$C_{n}=\frac{W_{ts}\times 10^{3}}{V_{air}}\times \frac{R_{ts}}{100}$$

*C_n_*: Theoretical concentration (unit: mg/L·min).

*W_ts_*: Amount of fib-CNFs solution used (unit: g).

*V_air_*: Supply flow rate of 16~20 L air per minute for 6 h (unit: L).

*R_ts_*: Concentration of aqueous fib-CNFs solution (%).

For a theoretical concentration of 7.0 mg/L, *W_ts_* is 8100 g, *V_air_* is 16 L, and *R_ts_* is 0.5%.$$\frac{7.0\;mg}{L\;\cdot \;min}=\frac{8100\times 1000\times 0.5}{16\times 60\times 6\times 100}$$

The test atmosphere was sampled by drawing it through a glass fiber filter (GB-100R, 55 mm diameter, Toyo Filter Paper Co., Ltd., Toyo, Japan). The mass of the evaporation residue collected on the filter was measured to determine the actual exposure concentrations. The GB-100R filter has a nominal pore size of 600 nm and a gas collection efficiency of 99.99%. Samples were collected at 1, 3, and 5 h after the start of exposure, with a collection flow rate of 1 L/min. The target concentrations for the fibrous cellulose nanofibers (fib-CNFs) were set at 1.5 mg/m^3^, 7.0 mg/m^3^, and 35 mg/m^3^, respectively (Table 2).

To eliminate residual moisture, laboratory air was drawn through the collected glass fiber filter at a flow rate of 3 L/min for 30 min. Subsequently, the actual exposure concentrations were calculated using the following equation:$$C_{Exp}=\frac{W_{crv}\times 10^{3}}{V_{cair}}$$

*C_Exp_*: Exposure concentration (unit: 0.01 mg/m^3^).

*W_crv_*: Atmospheric residue mass for collection test (minimum unit: 0.01 mg).

*V_cair_*: Collected air volume (minimum unit: 0.1 L).

To determine the actual exposure concentration of fibrous cellulose nanofibers (fib-CNFs) in the inhalation study, the following calculation was performed:

Assuming a collected fib-CNF mass of 0.25 mg on the glass fiber filter, with a collection duration of 6 min and an air sampling volume of 6 L, and accounting for an evaporation residue mass of 0.0058 mg, the actual exposure concentration is calculated to be 7.18 mg/L.

#### 2.3.5. Measurement of Exposed Particle Distribution

In the groups exposed to the test substance, the particle size distribution of aerosolized particles in the test atmosphere was measured. The measurements were conducted using a cascade impactor (01-130JDS, 8-stage classifier, InTox Products, Clinton, MS, USA) and a suction pump (Mini-pump with integrated flowmeter, MP-Σ300, Shibata Kagaku Co., Ltd., Shibata, Japan). The measurements were taken once a week for 1 h after the start of exposure, with a collection flow rate of 1 L/min. The mist containing fib-CNFs in the air had concentrations of 0.3, 1.4, and 7.0 mg/L, with the fib-CNFs solids constituting 0.5% of the mist. For clarity, these values are easier to interpret when expressed in cubic meters rather than liters. Therefore, the fib-CNFs content in 1 cubic meter of air corresponds to 0.3, 1.4, and 7.0 mg/L of mist, which will henceforth be referred to as 1.5, 7.0, and 35 mg/m^3^ of fib-CNFs (solid mass).

The collection times adhered to the schedule outlined in Table 3a. For additional verification, supplementary measurements were conducted after the exposure period under the conditions specified in Table 3b. The analytical balance utilized for these measurements has a precision of 0.01 mg. Shorter collection times result in minimal particulate matter accumulation on the filter, leading to decreased measurement accuracy. Conversely, excessively prolonged collection periods cause the accumulation of fib-CNFs on the filter, leading to clogging and hindering accurate collection. The collection durations specified in Table 2 and Table 3 were determined based on preliminary experiments.

The mass of the collected exposure samples was determined by weighing the collection plate before and after sampling (in milligrams). For confirmation collections taken after the exposure period, the collection plate was dried in a drying oven, and the mass of the collected exposure samples was calculated from the mass difference.

The parameters used for calculating the particle size distribution are as follows, with the Probit approximation method applied for calculations:(a)Calculate the particle size distribution parameters from the mass of the collected exposure samples and the effective cutoff diameter (ECD) of each stage (in nanometers).(b)Calculate the mass median aerodynamic diameter (MMAD) (minimum unit: 100 nm).(c)Geometric standard deviation (GSD) (minimum unit: 0.1).(d)Mass ratio of particles with a diameter of 4000 nm or less, which are considered to be inhalable by rats (minimum unit: 0.1%).

The Probit approximation method is commonly used in biostatistics to analyze binary response data and estimate the probability of an event occurring. In the context of particle size distribution, it can be applied to model the cumulative distribution function of particle sizes, facilitating the calculation of parameters such as MMAD and GSD.

### 2.4. Urinalysis

In the fourth week of the study (males, days 24 to 25; females, days 23 to 24), fresh urine samples were collected and analyzed. The parameters and methods used for the analysis are detailed in Table 4.

### 2.5. Hematological Tests

The examination parameters and methods are detailed in Table 5. For the assessment of Prothrombin Time (PT) and Activated Partial Thromboplastin Time (APTT), 0.6 mL of whole blood was collected into a blue-top tube containing 3.2% sodium citrate as an anticoagulant. The sample was then centrifuged at 12,000× *g* for 3 min at 4 °C to separate the plasma.

### 2.6. Blood Biochemical Tests

After allowing approximately 2 mL of whole blood to clot at room temperature for 30 to 60 min, serum was separated by centrifugation at 1500× *g* for 10 min at 4 °C. The analytical parameters and methodologies are detailed in Table 6. The serum samples were then stored at approximately –80 °C (–60 °C or lower) until further analysis.

### 2.7. Pathological Examination

Autopsies were performed on the 29th day. After blood collection, the animals were euthanized by exsanguination from the abdominal aorta, followed by organ examinations. The examined organs included the heart*, aorta, lymph nodes, thymus*, spleen*, femur and bone marrow, sternum, nasal cavity, pharynx, trachea, lungs/bronchi*, tongue, esophagus, stomach, duodenum, jejunum, ileum, appendix, colon, rectum, salivary gland*, liver*, pancreas, kidneys*, bladder, testis*, epididymis*, seminal vesicle/epididymis, prostate, ovary*, uterus*, vagina, pituitary gland*, thyroid/parathyroid glands*, adrenal glands*, brain*, olfactory bulb, cerebrospinal fluid, eyes/optic nerves/Harderian gland, skeletal muscle/sciatic nerve, skin/mammary gland, and any other organs/tissues with gross abnormalities. Organ weights were measured for those marked with an asterisk (*).

### 2.8. Bronchoalveolar Lavage Fluid (BALF) Examination

On day 29, bronchoalveolar lavage (BAL) was performed on the right lung, including the trachea and bronchi, during dissection, while the left lung’s bronchi were clamped. Phosphate-buffered saline (PBS, Cat. No. T9181: Takara Bio Inc., Chuo City, Japan) was utilized to process and recover the BAL fluid (BALF). Cell counts in the BALF were determined using an automated hematology analyzer (XT-2000iV: Sysmex Corporation, Japan), which employs a semiconductor laser for flow cytometry to enumerate total white blood cells, macrophages, neutrophils, lymphocytes, and eosinophils, providing both ratios and absolute counts. For the macrophage phagocytosis assay, the BALF supernatant cell suspension was diluted with PBS (-), containing 0.1% bovine serum albumin (BSA) and 0.05 mM ethylenediaminetetraacetic acid (EDTA-2K). A cytospin (Cytospin 3: SHANDON) was employed to prepare smears for staining and observation. The BALF examination was conducted on day 29, coinciding with the dissection, and was performed immediately on the same day. BAL is a well-established technique in mice, rats, and humans; however, it requires a high level of skill. Most cells collected in the BALF are macrophages, and it is advisable to measure them within one hour of collection, so we adhered to this procedure [29].

### 2.9. Statistical Analysis

Data analysis was performed using the tsPharma LabSite safety testing system (Fujitsu Limited, Minato City, Japan). To assess the homogeneity of variances, Bartlett’s test was conducted on the quantitative data at a significance level of 5%. If the variances were homogeneous, multiple comparisons were made using Dunnett’s test. If the variances were heterogeneous, Steel’s test was applied for multiple comparisons, with significance levels set at 5% and 1% for two-sided tests.

Bartlett’s test evaluates the null hypothesis that multiple groups have equal variances. If the test indicates significant differences, it suggests that the assumption of equal variances may not hold, which can affect the validity of parametric tests like ANOVA. Dunnett’s test is used to compare multiple treatment groups to a single control group, adjusting for multiple comparisons to control the Type I error rate. Steel’s test is a non-parametric method for comparing multiple groups to a control group when variances are unequal, providing an alternative when assumptions for parametric tests are not met.

## 3. Results

### 3.1. Fib-CNFs Fiber Length and Diameter

The measured lengths and diameters of the nanofibers are shown in Figure 2. The mean nanofiber length (Figure 2a) was 624 nm (standard deviation: 199 nm), and the mean diameter (Figure 2b) was 42.9 nm (standard deviation: 9.2 nm), as measured by transmission electron microscopy (TEM) and field-emission scanning electron microscopy (FE-SEM). The average diameter closely matched the specifications provided by the manufacturer. However, the actual fiber lengths might be longer than indicated in Figure 2. This discrepancy arises because cellulose nanofibrils (CNFs) tend to entangle and form spherical aggregates, and measurements were taken only from fibers separated from these aggregates (Figure 2c).

In TEM observations (Figure 2c), only the crystalline regions of cellulose appeared as needle-like structures. As a result, fiber diameters could not be accurately measured using TEM alone, and were instead determined using FE-SEM (Figure 2d).

### 3.2. Particle Size Distribution

The particle size distribution measurements are summarized in Table 7. The cascade impactor, composed of multiple inertial impactor stages, separates particles based on size. With each successive stage, the cut-off diameter decreases, allowing larger particles to be captured at the upper stages. However, measurements taken during the exposure period were deemed unreliable because a substantial amount of fibrous cellulose nanofibrils (fib-CNFs) leaked from the stainless steel plate of the cascade impactor after being sprayed as an aerosol mist.

As shown in Table 7, measurements taken under stable conditions after the exposure period revealed a mass median aerodynamic diameter (MMAD) ranging from 600 to 1000 nm, with a geometric standard deviation (GSD) of 2.0 to 2.4. Furthermore, 96.0% to 99.7% of the particles had diameters of 4000 nm or less, a size range considered respirable for rats. These results were deemed reliable based on the total mass of the aerosolized test substance collected by the cascade impactor.

Figure 3 displays a scanning electron microscopy (SEM) image of fib-CNFs collected on a filter (exposure concentration: 35 mg/m^3^). The red circle in the image represents a diameter of 1000 nm, corresponding to the MMAD. From this, it can be inferred that fib-CNFs entering the lungs are already present as aggregates rather than individual needle-like structures at the point of inhalation by rats.

### 3.3. General Condition of Animals

No exposure-related changes attributable to the test substance were observed. However, crust formation was noted in male rats, likely as a nonspecific effect associated with retention within the nasal exposure apparatus.

### 3.4. Body Weight

Body weight data are presented in Figure 4. Throughout the study, no statistically significant differences in body weight were observed between male and female rats in the test substance exposure groups compared to the control groups.

### 3.5. Food Intake

Food intake data indicate no statistically significant differences between male and female rats in the test substance exposure groups compared to the control groups throughout the study period.

### 3.6. Water Intake

Water intake measurements showed a statistically significant increase on day 15 in female rats exposed to 35 mg/m^3^ of the test substance compared to the control group. However, this was considered a transient variation and was deemed to have no toxicological relevance.

### 3.7. Ophthalmological Examination

Ophthalmological examinations identified scattered cases of granular opacities in the crystalline lens in both male and female rats during the exposure period, compared to baseline levels prior to exposure. These findings were deemed incidental and unrelated to the test substance exposure.

### 3.8. Urinalysis

Urinalysis results revealed no significant differences between the test substance exposure groups and the control groups. However, statistically significant increases in potassium levels in females and chloride levels in males were observed in the 1.5 mg/m^3^ exposure group compared to the control group (Appendix A). As these changes were not dose-dependent, they were interpreted as random variations with no toxicological relevance.

### 3.9. Hematological Examination

Hematological examination results are shown in Figure 5. In female rats exposed to 35 mg/m^3^ of the test substance, statistically significant increases were observed in mean corpuscular volume (MCV), mean corpuscular hemoglobin (MCH), the neutrophil ratio, and absolute neutrophil count, alongside a decrease in the lymphocyte ratio compared to the control group. However, the elevated MCV and MCH values were within the mean ± 2SD range of the test facility’s reference values. Additionally, the lower lymphocyte ratio was interpreted as a relative change caused by the increased neutrophil ratio and count. These variations were therefore deemed to have no toxicological relevance.

### 3.10. Blood Biochemical Tests

Blood biochemical test results are presented in Figure 6. Statistically significant findings included decreased aspartate aminotransferase (ASAT) and alanine aminotransferase (ALAT) levels in females exposed to 1.5 mg/m^3^, elevated total cholesterol levels in males exposed to 7.0 mg/m^3^, and increased phospholipid levels in males exposed to 7.0 mg/m^3^ and 35 mg/m^3^.

There was no association between the reduced ASAT and ALAT levels and the increased total cholesterol levels or exposure concentrations. Furthermore, the elevated phospholipid levels remained within the mean ±2 standard deviations of the facility’s background data. These variations were therefore deemed random fluctuations with no toxicological relevance.

### 3.11. Organ Weights

Organ weight measurement results are presented in Figure 7. Statistically significant increases in both absolute and relative lung weights were observed in male and female rats exposed to 7.0 mg/m^3^ and 35 mg/m^3^, compared to the control group. No significant changes were observed in other organs, suggesting that the inhaled fibrous cellulose nanofibrils (fib-CNFs) primarily accumulated in the lungs.

It is hypothesized that the increased lung weight reflects the cumulative effect of the fib-CNFs inhaled by the rats over the 28-day exposure period, along with the associated proliferation of macrophages and other immune cells in response to the fibers.

### 3.12. Autopsy Findings

No changes attributable to the test substance were observed. Nonspecific findings, such as crust formation due to the retention and unilateral atrophy of the testes and epididymides, were considered to be spontaneous in nature.

Bronchoalveolar lavage fluid (BALF) test results are shown in Figure 8 and Appendix B. In males exposed to 1.5 mg/m^3^, there was a statistically significant increase in total cell count, a higher lymphocyte-to-total cell ratio, and elevated absolute numbers of lymphocytes, neutrophils, eosinophils, and macrophages in the BALF. In contrast, females in the 1.5 mg/m^3^ exposure group exhibited a lower neutrophil-to-total cell ratio and a decrease in the absolute number of neutrophils. Additionally, females had a lower macrophage-to-total cell ratio and a higher absolute number of lymphocytes.

In the 7 mg/m^3^ exposure group, males exhibited an increase in total cell count, elevated lymphocyte and eosinophil ratios, and higher absolute numbers of lymphocytes, neutrophils, eosinophils, and macrophages in the bronchoalveolar lavage fluid (BALF). However, the macrophage-to-total cell ratio decreased. Females in the 7 mg/m^3^ exposure group showed increased total cell count, lymphocyte count, neutrophil count, and eosinophil count. The neutrophil percentage of total cells and the absolute number of neutrophils increased, as did the absolute number of macrophages, while the macrophage-to-total cell ratio decreased.

In the 35 mg/m^3^ exposure group, males exhibited an increase in total cell count, as well as elevated percentages and absolute numbers of lymphocytes, neutrophils, and eosinophils in the bronchoalveolar lavage fluid (BALF). The absolute number of macrophages also increased, while the macrophage-to-total cell ratio decreased. Similarly, females in the 35 mg/m^3^ exposure group showed elevated percentages of neutrophils and eosinophils, along with absolute increases in neutrophils and eosinophils. The absolute numbers of lymphocytes and macrophages also increased, but the macrophage-to-total cell ratio was lower. Importantly, this decrease in the macrophage ratio was considered to have no toxicological significance, as it correlated with the very high absolute numbers of lymphocytes, neutrophils, and eosinophils.

Notably, the phagocytosis of fibrous cellulose nanofibrils (fib-CNFs) by macrophages was observed in both male and female rats exposed to 7 mg/m^3^ and 35 mg/m^3^ of fib-CNFs. The presence of arrow-shaped structures (Figure 8c,g,h) that appear bright under polarized light, along with the inability to observe individual fibers of the fib-CNFs under an optical microscope, suggest that these structures are aggregates of CNFs. The bronchoalveolar lavage fluid (BALF) test results indicate no significant differences in the amounts of CNF aggregates phagocytosed by macrophages between males and females. Additionally, CNF aggregates could not be detected under an optical microscope at 1.5 mg/m^3^, but were confirmed at 7 mg/m^3^ and 35 mg/m^3^. However, no differences were observed between the exposure concentrations.

### 3.13. Histopathological Analysis

Figure 9, Figure 10, Figure 11 and Figure 12 display microscopic images of lung and lymph node tissues. Histopathological examination revealed that inhalation exposure to cellulose nanofibrils (CNFs) induced tissue changes in the lungs and peribronchial lymph nodes of both male and female rats, as summarized in Table 8.

In the lungs, the dose-dependent infiltration of foamy alveolar macrophages was observed at concentrations of 1.5 mg/m^3^ and higher in both males and females, accompanied by mild infiltration of inflammatory cells. In the peribronchial lymph nodes, clusters of foamy macrophages were observed in females exposed to 1.5 mg/m^3^ and higher, and in males exposed to 7 mg/m^3^ and higher. However, this change was noted in only one female in the 1.5 mg/m^3^ group, and no significant differences were observed between the sexes in the 7 mg/m^3^ and 35 mg/m^3^ exposure groups.

Various nonspecific tissue changes were also observed in all groups, including the control group. Since these changes are commonly observed in rats and showed no clear differences between groups, they were considered unrelated to the test substance.

As mentioned earlier, the dose-dependent infiltration of foamy alveolar macrophages, along with the mild infiltration of inflammatory cells, was observed at concentrations of 1.5 mg/m^3^ and higher in both males and females. Additionally, the aggregation of foamy macrophages occurred in the bronchial lymph nodes of females exposed to 1.5 mg/m^3^ and higher, and in males exposed to 7 mg/m^3^ and higher. Alveolar macrophage infiltration and the accumulation of macrophages in the peribronchial lymph nodes are common responses following inhalation exposure to particulate matter or the intratracheal administration of such matter. Various inflammatory responses are known to occur depending on the properties and quantity of the inhaled substance.

The changes observed in this study are believed to be related to exposure to the test substance. Based on these findings, histopathological changes due to CNF inhalation were observed in both male and female rats exposed to 1.5 mg/m^3^. Therefore, these findings suggest that the exposure concentration below which no toxicological effects (NOAEL) occur should be considered to be less than 1.5 mg/m^3^.

## 4. Discussion

Fibrous cellulose nanofibrils (fib-CNFs) are considered the standard for cellulose nanofibers, and it is crucial for both manufacturers and consumers to determine whether inhalation of these fibers poses a risk of lung cancer or mesothelioma, or if such risks are absent. In the 28-day continuous inhalation exposure test conducted on rats, the degree of inflammation caused by fib-CNFs deposited in the alveoli or terminal bronchioles was minimal, and there was no weight loss, no health abnormalities, and no significant changes in blood or biochemical tests. However, evidence was provided that macrophages phagocytose fib-CNFs or their aggregates as foreign particles, supported by an increase in lung weight, as well as the histological examination of the alveoli and the bronchoalveolar lavage fluid (BALF) analysis. It was also confirmed that some of the fib-CNFs inhaled into the lungs were able to traverse the pleura and move to the mediastinal lymph nodes via the thoracic lymphatic vessels. Notably, the aggregation of macrophages in the alveoli caused by fib-CNFs was observed, but alveolar hyperplasia, a hallmark of carcinogenic substances, was not seen. This suggests that the macrophage aggregation caused by fib-CNFs resembles the inflammation induced by general fine particles, indicating an extremely low potential for carcinogenesis.

There was little to no difference between the sexes. As the inhalation exposure of CNFs (Figure 5j) increased, the number of neutrophils in the blood rose in both males and females. Examining the BALF test results in Figure 8 and the absolute neutrophil count in the BALF tests for both males and females (Appendix B), no clear sex differences were observed. Although a statistically significant difference appears to exist, further verification with a larger sample size is necessary.

The increase in lung weight shown in Figure 7 was similar in both males and females, but histopathological examination of the lungs (Table 7) revealed that macrophage aggregation was more pronounced in females than expected. Alveolar foam macrophage infiltration was observed in a dose-dependent manner in both males and females at doses of 1.5 mg/m^3^ or higher, accompanied by the mild infiltration of inflammatory cells. Aggregates of foam macrophages were also seen in the peribronchial lymph nodes of female rats in the 1.5 mg/m^3^ group and male rats in the 7.0 mg/m^3^ group. This change was observed in only one female rat in the 1.5 mg/m^3^ group, and no clear differences were found between the sexes in the 7.0 and 35 mg/m^3^ groups. Based on these results, it is suggested that the effects of fib-CNFs on the lungs may be slightly stronger in females.

Saleh et al. [30] examined the effects of the intratracheal instillation of multi-walled carbon nanotubes (MWCNTs) in rats, and found that when the total dose of fibers in the rat lung exceeded the lung burden of insoluble particles, such as carbon nanotubes (approximately 1–3 mg per gram of lung tissue), it could alter the retention kinetics within the lung. They proposed that alveolar hyperplasia lesions may or may not develop into tumors, but that cells that eventually become carcinogenic always first develop alveolar hyperplasia, followed by benign tumors (adenomas), and eventually malignant tumors (carcinomas).

Fujita et al. [23] administered TEMPO-oxidized cellulose nanofibers (CNFs), mechanically defibrinated CNFs, and short-fiber CNFs via a single intratracheal administration at a dose of approximately 0.52 mg of CNF per rat, which is less than half the dose used by Saleh et al. [30]. In our research, assuming the rats inhaled 1.4 mL of CNFs per breath and breathed 150 times per minute [31] for 6 h a day over 28 days, the total CNF inhaled at an exposure concentration of 35 mg/m^3^ would amount to 75.6 mg per body. Considering the lung weight of male rats is approximately 1.4 g, this means they inhaled a significantly larger amount than the single intratracheal dose administered by Fujita et al. We are continuing efforts to quantify the amount of CNFs present in the lung tissue.

Regarding the effects of CNF fiber length on lung tissue inflammation, Fujita et al. [22] reported that phosphorylated CNFs and TEMPO-oxidized CNFs, which were short and thin, reached the alveoli. In contrast, the fib-CNFs used by them were longer (1700 nm), so only some of the fibers reached the alveoli, leading to a lower degree of lung inflammation after intratracheal administration compared to phosphorylated CNFs and TEMPO-oxidized CNFs. However, the fib-CNFs used in our study had a fiber length of 624 nm, which allowed them to reach the alveoli.

We hypothesize that fib-CNFs exist in aqueous solutions as entangled aggregates, rather than as isolated fibers, due to mechanical defibrillation.

This is supported by the observation in Figure 3, which shows that when the aerosol containing fib-CNFs dries, the CNFs tend to aggregate easily. Consequently, in Figure 8, Figure 9, Figure 10, Figure 11 and Figure 12, we could not detect single fib-CNFs, only aggregates, leading us to speculate that CNFs, unlike asbestos or carbon nanotubes (CNTs), which are needle-like and can be found in the alveoli and macrophages, have a limited ability to stimulate cellular responses.

On the other hand, since CNFs are biological substances, distinguishing them from lung tissue using macroscopic observation and quantitative analysis is challenging. Moreover, CNFs are transparent to light, and only the crystalline components can be identified under polarized light microscopy. Thus, to specifically and sensitively detect CNFs in biological tissues, a novel detection method is required, and we are actively working on developing such a method.

In this study, we evaluated the safety of CNFs by replacing previous CNT studies with CNF-specific assessments, following OECD guidelines [26]. In a 90-day inhalation study of CNTs as per OECD-TG413 [32,33], mild acute to chronic inflammation of the alveolar septal walls and type II pneumocyte hypertrophy/hyperplasia were observed in rats exposed to 25 mg/m^3^ of CNTs. Additionally, the mild inflammation of the terminal bronchioles and alveolar ducts, along with some thickening of the septal walls and hypertrophy/hyperplasia of type II pneumocytes, was noted. Increased bronchoalveolar lavage (BAL) fluid recovery of polymorphonuclear cells (PMN), lactate dehydrogenase (LDH), alkaline phosphatase (ALKP), and metalloproteinase (MTP), along with enhanced cell proliferation, were also observed in these animals.

Tsuda et al. [34,35,36,37,38] also reported that CNTs can induce malignant mesothelioma and lung cancer. While the iron content of multi-walled CNTs (MWCNTs) is not directly linked to mesothelial tumor induction in the abdominal cavity, it may play a role in MWCNT-mediated carcinogenesis in the lungs. Iron can potentially generate oxygen radicals via Fenton reactions, even in the absence of fiber interactions with macrophages. Furthermore, the surface area of nanomaterials could significantly contribute to the carcinogenic potential of MWCNT-B fibers wrapped in granulation tissue. This is because the surface area of MWCNT-B is approximately 20 times larger than that of MWCNT-A for the same mass of material. Thus, the larger surface area of MWCNT-B may enhance its pulmonary carcinogenicity.

The specific surface area of CNFs is comparable to that of CNTs, though CNFs are unlikely to contain iron like CNTs, and as such, their carcinogenic potential is presumed to be very low. Moving forward, it will be important to reaffirm that macrophage aggregation induced by CNFs does not progress to alveolar hyperplasia.

## 5. Conclusions

The inhalation of mechanically defibrated cellulose nanofibers (fib-CNFs) significantly increased both the total cell count in the bronchoalveolar lavage fluid (BALF) and the lung weight. These changes are thought to be related to inflammatory processes and the recruitment of lymphocytes, neutrophils, eosinophils, and macrophages into the lungs, which contribute to the increases in lung weight due to the inhalation of particulate matter.

In contrast, no significant differences were observed in body weight, or in hematological or biochemical parameters, when compared to the control group. Importantly, lung inflammation was observed in both male and female rats at the lowest exposure concentration of 1.5 mg/m^3^, which corresponds to a 0.5% fib-CNF solution. Based on these findings, it was concluded that the threshold for safe exposure without inducing inflammation (NOAEL) is 1.5 mg/m^3^ or lower.

### Ethical Approval

The University of Fukui Animal Experiment Regulations are as follows: University of Fukui Regulation No. 2 of 10 January 2007 Life science research involving animal experiments at universities and other institutions is a necessary means not only for the development and deployment of human health, welfare, and advanced medical care, but also for the advancement of research fields such as animal health promotion. These regulations are based on the “Law Concerning the Welfare and Management of Animals” (Law No. 105 of 1973, hereinafter referred to as the “Law”) and the “Care and Use of Laboratory Animals” guidelines. The “Standards for the Care and Keeping of Laboratory Animals and the Reduction of Pain and Suffering” (Ministry of the Environment Notification No. 88 of 2006, hereinafter referred to as the “Care and Keeping Standards”) and the “Basic Guidelines for the Conduct of Animal Experiments in Research Institutions, etc.” (Ministry of Education, Culture, Sports, Science and Technology Notification No. 71 of 2006, hereinafter referred to as the “Basic Guidelines”) are also incorporated. The Guidelines for the “Appropriate Conduct of Animal Experiments” (Science Council of Japan, 1 June 2006, hereinafter referred to as the “Guidelines”) provide a framework for the implementation of animal experiments considering scientific perspectives, animal welfare, environmental conservation, and the safety of faculty members, students, and others involves in animal experiments. The Ethics Committee for Animal Experiments for this study was held on 24 October 2022.

The animal experiments were conducted in accordance with the “Law Concerning the Welfare and Management of Animals” (Law No. 39, 201196), the “Standards for the Care and Keeping of Laboratory Animals and the Reduction of Pain” (Ministry of the Environment Notification No. 84, September 2013), the “Guidelines for the Proper Conduct of Animal Experiments” (Science Council of Japan, June 2006), and the “Guidelines for the Proper Implementation of Animal Experiments” (Ministry of Health, Labor and Welfare and Ministry of Agriculture, Forestry and Fisheries of Japan, June 2006).

This research was done with the University of Fukui (approval number R06077) and the Mediford Corporation (approval number B220613) Animal Experiment Guidelines based on these Basic Guidelines for the Conduct of Animal Experiments established by the Ministry of Health, Labour and Welfare and the Ministry of Agriculture, Forestry and Fisheries of Japan.

This study was not subject to Good Laboratory Practice (GLP), but various operations and data handling processes were conducted in accordance with the standard operating procedures (SOPs) of the facilities wherein the study was conducted.

This study was conducted in accordance with OECD Guidelines for the Testing of Chemicals, Section 4, 27 June 2018, as specified in the guideline: “OECD Guidelines for the Testing of Chemicals, TG412 (28-day (subacute) inhalation toxicity test)”.

## Figures and Tables

**Figure 1 nanomaterials-15-00214-f001:**
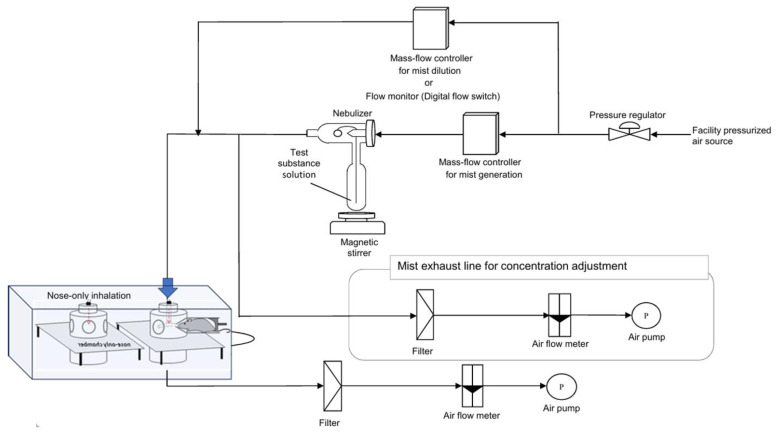
Schematic diagram of nasal exposure device. The test substance solution with CNFs is connected to a two-fluid nebulizer, and compressed air is supplied to generate a mist of the exposed CNFs, which is then exposed to the animals.

**Figure 2 nanomaterials-15-00214-f002:**
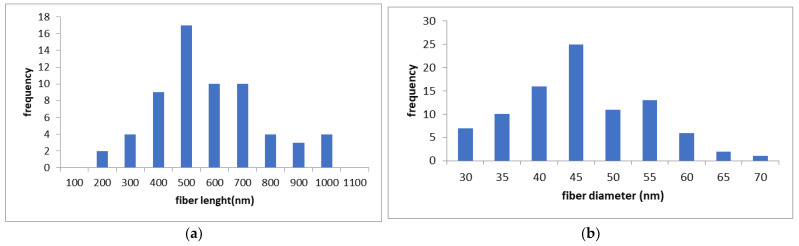
Fib-CNFs (BiNFi-s ultra-short FMa1002): fiber length and diameter; (**a**) fiber length distribution, (**b**) fiber diameter distribution, (**c**) TEM image ×10,000, (**d**) FE-SEM image ×50,000.

**Figure 3 nanomaterials-15-00214-f003:**
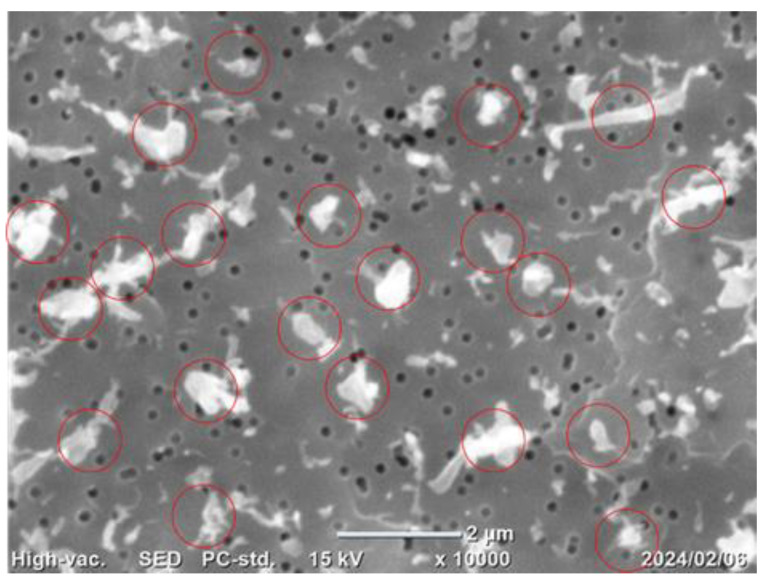
SEM photograph of fib-CNFs collected on a filter (exposure concentration: 35 mg/m^3^). The diameter of the red circle is 1000 nm. ×10,000.

**Figure 4 nanomaterials-15-00214-f004:**
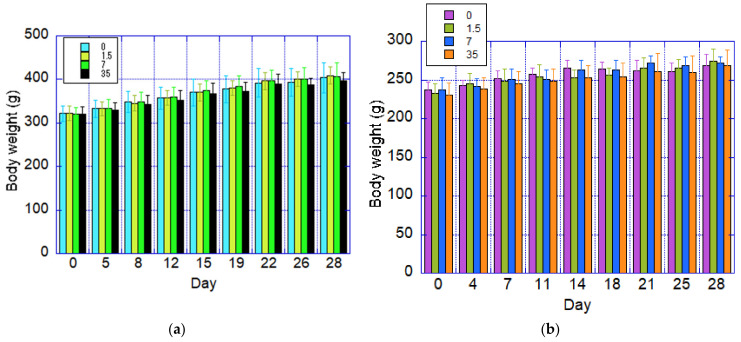
Changes in body weight in animals that inhaled CNFs for 28 days. Dose amounts are shown in the upper left of each figure. Unit is mg/m^3^. (**a**) Males, (**b**) females.

**Figure 5 nanomaterials-15-00214-f005:**
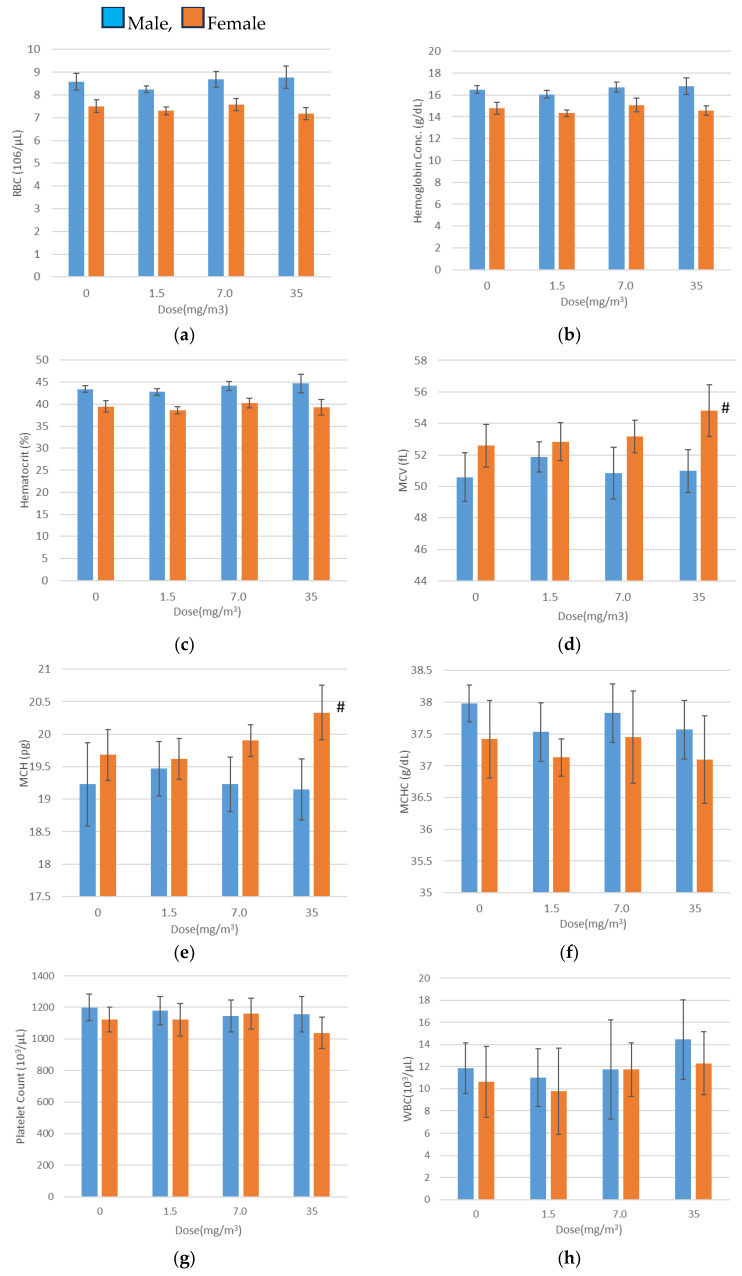
Hematology (mean ± S.D.) Day 29. (**a**) RBC, (**b**) Hemoglobin concentration (**c**) Hematocrit, (**d**) MCV, (**e**) MCH, (**f**) MCHC, (**g**) Platelet count, (**h**) WBC, (**i**) Lymphocyte, (**j**) Neutrophil. Significantly different from the control group; #, *p* < 0.05; ##, *p* < 0.01 (Dunnett test).

**Figure 6 nanomaterials-15-00214-f006:**
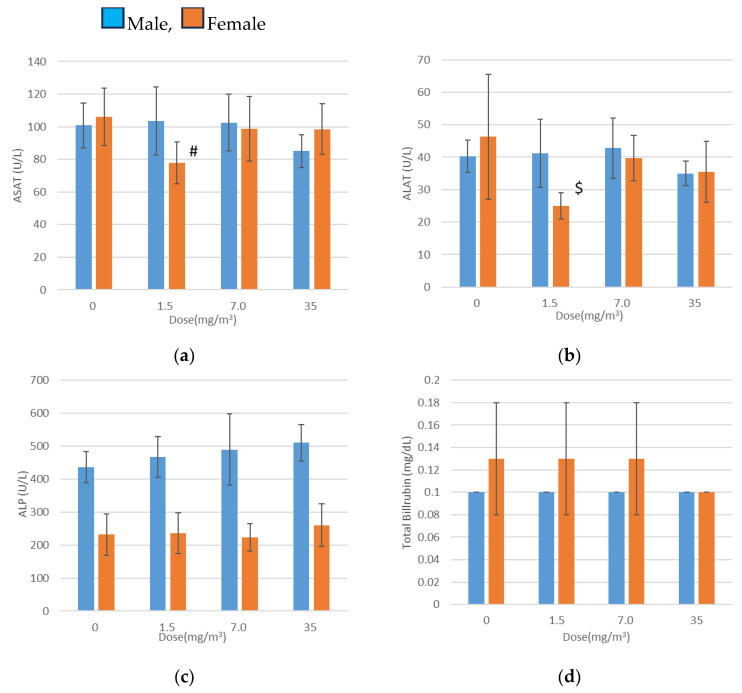
Blood chemistry (mean ± S.D.) Day 29. (**a**) ASAT, (**b**) ALAT, (**c**) ALP, (**d**) Total Bilirubin, (**e**) Urea nitrogen, (**f**) CRE, (**g**) Glucose, (**h**) Total cholesterol, (**i**) Phospholipid, (**j**) Triglyceride, (**k**) Total protein, (**l**) A/G Ratio, (**m**) Albumin, (**n**) α1 Globulin, (**o**) α2 Globulin, (**p**) β Globulin, (**q**) *γ* Globulin, (**r**) Ca, (**s**) Inorganic phosphorus, (**t**) Na, (**u**) K, (**v**) Cl. Significantly different from the control group; #, *p* < 0.05; ##, *p* < 0.01 (Dunenett test). Significantly different from the control group; $, *p* < 0.05 (steel test).

**Figure 7 nanomaterials-15-00214-f007:**
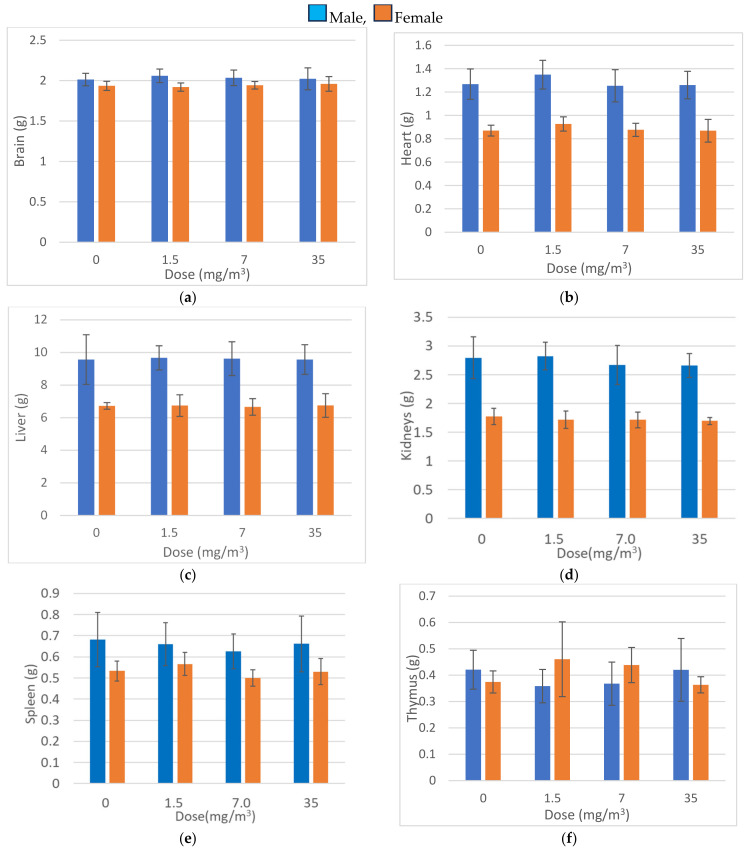
Organ weights (mean ± S.D.) Day 29. (**a**) Brain, (**b**) Heart, (**c**) Liver, (**d**) Kidneys, (**e**) Spleen, (**f**) Thymus, (**g**) Pituitary, (**h**) Thyroids, (**i**) Adrenals, (**j**) Lungs, (**k**) Submandibular Glands, (**l**) Testes(L) Epididymides(R), (**m**) Ovaries(L) Uterus(R). Significantly different from the control group; ##, *p* < 0.01 (Dunnett test). Significantly different from the control group; $, *p* < 0.05 (steel test).

**Figure 8 nanomaterials-15-00214-f008:**
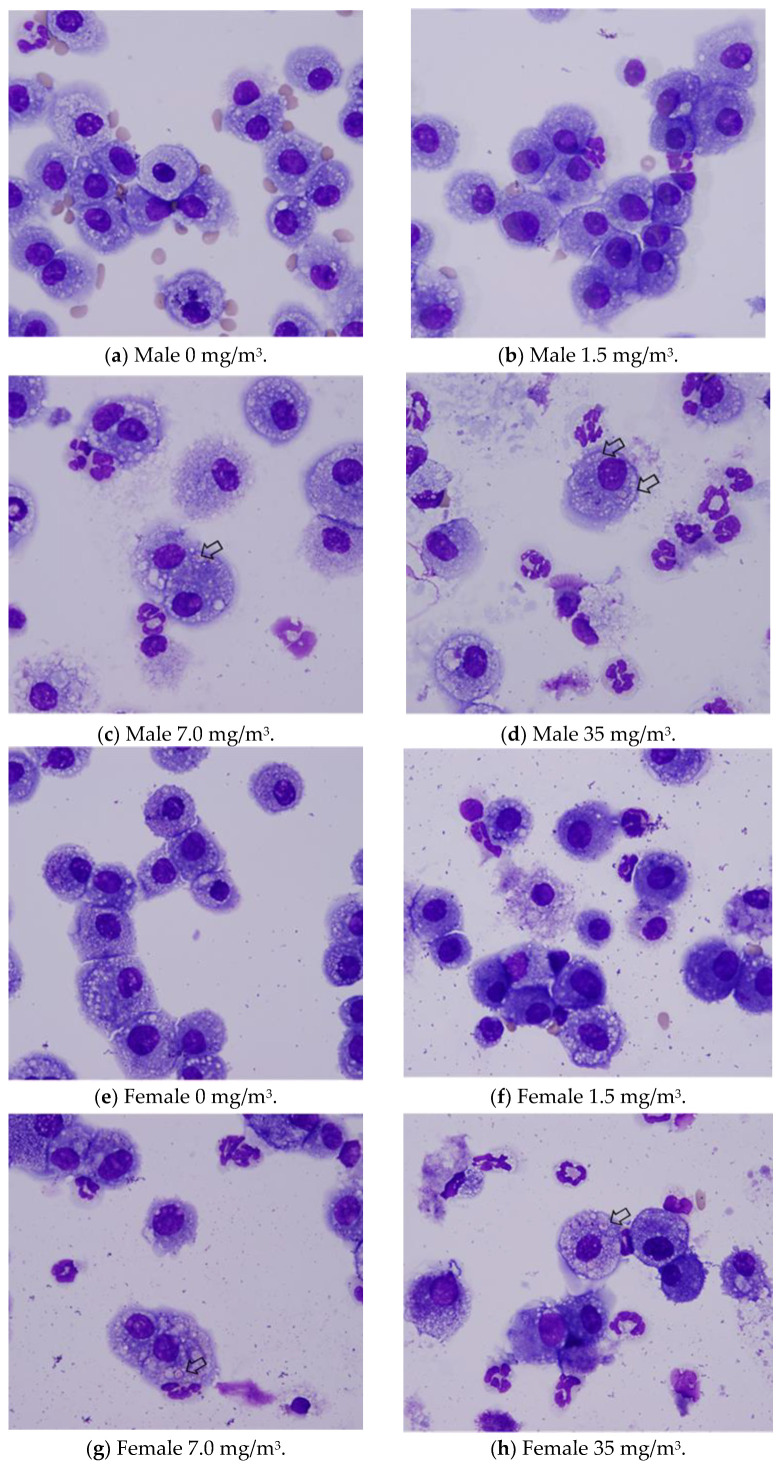
BALF (bronchoalveolar lavage fluid) of male and female rats. Arrows indicate macrophage phagocytosis. CNFs phagocytosed by macrophages are observed as aggregates, not as needles or fibers, ×40.

**Figure 9 nanomaterials-15-00214-f009:**
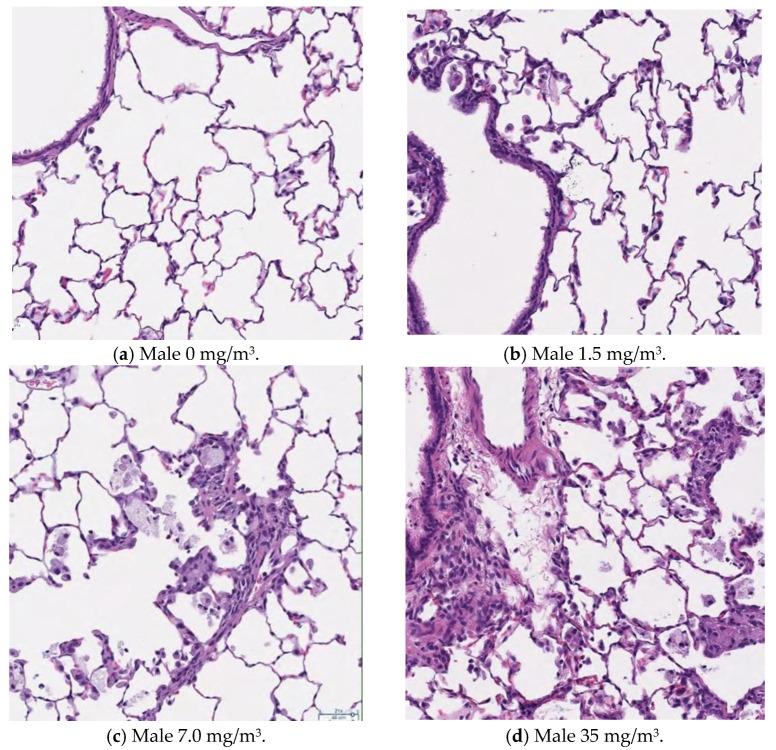
Pathological analysis and cellular histology of male rat alveoli. Aggregation of macrophages becomes more pronounced as the exposure to CNFs increases, ×20.

**Figure 10 nanomaterials-15-00214-f010:**
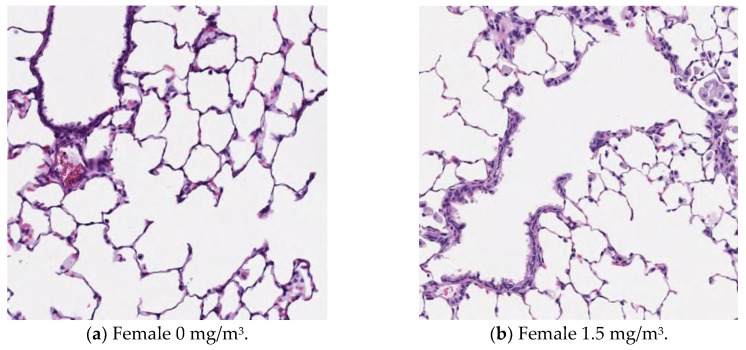
Pathological analysis of the alveoli of female rats. Aggregation of macrophages becomes more pronounced as the exposure to CNFs increases, ×20.

**Figure 11 nanomaterials-15-00214-f011:**
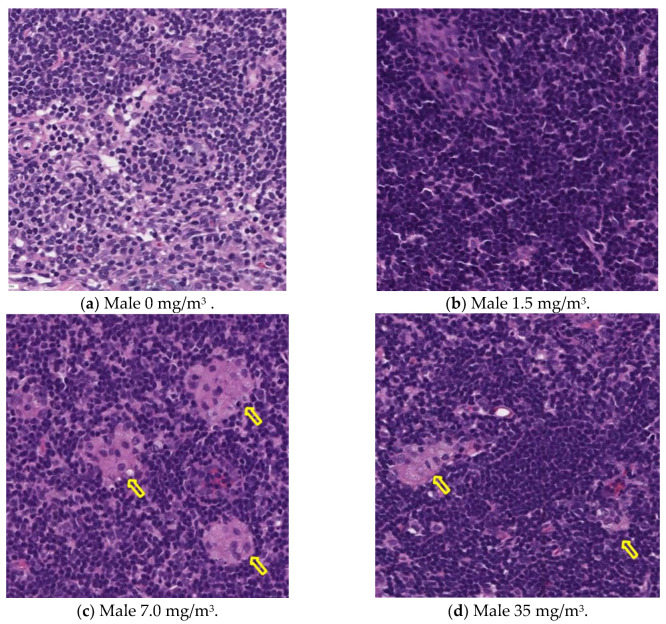
Pathological analysis of the lymph nodes in male rats. Macrophage aggregation (yellow arrows) can be seen in the lymph nodes of the groups exposed to 7.0 mg/m^3^ and 35 mg/m^3^ of CNFs, ×20.

**Figure 12 nanomaterials-15-00214-f012:**
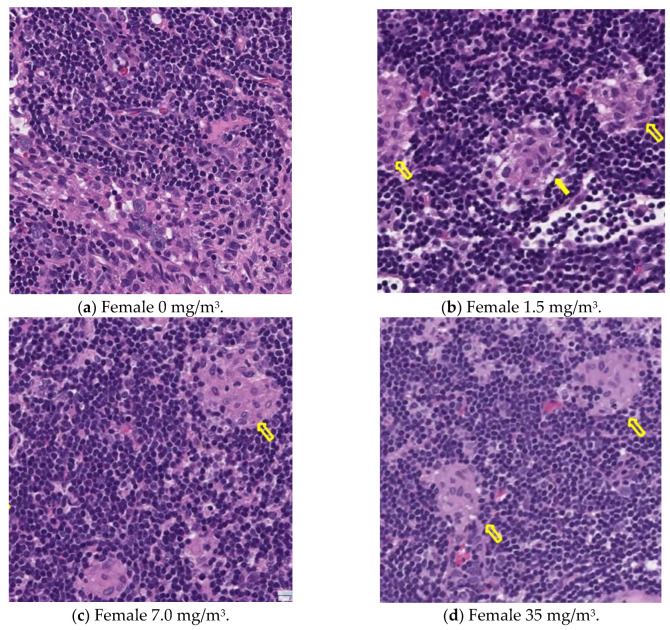
Pathological analysis of the lymph nodes in female rats. Macrophage aggregation (yellow arrows) in the lymph nodes were observed in all groups exposed to CNFs, ×20.

**Table 1 nanomaterials-15-00214-t001:** Group composition of rats.

fib-CNFs	Exposure Concentration (mg/m^3^)	Number of Animals (Animal Number)			
		Male		Female	
Purified water (control group)	0	6	(10,101–10,106)	6	(50,101–50,106)
fib-CNFs	1.5	6	(10,201–10,206)	6	(50,201–50,206)
fib-CNFs	7.0	6	(10,301–10,306)	6	(50,301–50,306)
fib-CNFs	35	6	(10,401–10,406)	6	(50,401–50,406)

**Table 2 nanomaterials-15-00214-t002:** Theoretical exposure concentration vs. collection time.

Theoretical exposure concentration (mg/m^3^)	1.5	7.0	35
Collection time (min)	120	30	6

**Table 3 nanomaterials-15-00214-t003:** (**a**) Collection at 1 h after the start of exposure. (**b**) Collection for confirmation after the end of exposure.

(**a**)			
Exposure concentration (mg/m^3^)	1.5	7.0	35
Collection time (min)	100	30	5
(**b**)			
Exposure concentration (mg/m^3^)	1.5	7.0	35
Collection time (min)	300	300	100

**Table 4 nanomaterials-15-00214-t004:** Items and methods of urinalysis.

Item (Abbreviation, Unit)	Method
pH	Dipstick test
Protein	Dipstick test
glucose	Dipstick test
Ketone body	Dipstick test
Bilirubin	Dipstick test
Occult blood	Dipstick test
Urinary Sediment	Specimen stained with Sternheimer–Malbin Centrifugation conditions: 500× *g*, 5 min, room temperature
Specific gravity	Refraction method
Urine volume (mL)	Measured with a measuring cylinder
Creatinine (mg/dL)	Enzymatic method (Creatininase-POD method)
Sodium (Na, mmol)	Ion selective electrode method
Potassium (K, mmol)	Ion selective electrode method
Chlor (Cl, mmol)	Ion selective electrode method

MARTISTICS: Siemens Healthcare Diagnostics K.K. (Measuring equipment). Test paper method: Clinitec Advantus (Siemens Healthcare Diagnostics K.K.). Specific gravity: Uricon-JE (Atago Corporation). Electrolytes and creatinine: TBA-2000FR (Canon Medical Systems Inc.). Electrolytes were creatinine-corrected values (mol/g-Cre).

**Table 5 nanomaterials-15-00214-t005:** Blood test items and methods of measurement.

Item (Abbreviation, Unit)	Method
Red blood cell count (RBC, ×10^6^/μL)	Sheath flow DC detection method
Hemoglobin concentration (HGB, g/dL)	SLS-hemoglobin method
Hematocrit level (HCT, %)	Erythrocyte pulse wave height detection method
Mean corpuscular volume (MCV, fL)	Calculated from RBC and HCT
Mean corpuscular hemoglobin (MCH, pg)	Calculated from RBC and HGB
Mean corpuscular hemoglobin concentration (MCHC, g/dL)	Calculated from HGB and HCT
Platelet count (PLT, ×10^3^/μL)	Sheath flow DC detection method
Reticulocyte fraction (%)	Flow cytometry method using semiconductor laser
Prothrombin time (PT, s)	Coagulation method
Activated partial thromboplastin time	Coagulation method
Activated partial thromboplastin time (APTT, s)	Flow cytometry using semiconductor laser
White blood cell count (WBC, ×10^3^/μL)	Flow cytometry using semiconductor laser

(Measuring equipment) PT and APTT: CN-3000 (Sysmex Corporation, Kobe, Japan). Others: XT-2000iV (Sysmex Corporation).

**Table 6 nanomaterials-15-00214-t006:** Items and methods of blood biochemical tests.

Item (Abbreviation, Unit)	Method
ASAT (GOT, U/L)	UV-rate method (JSCC improved method)
ALAT (GPT, U/L)	UV-rate method (JSCC improved method)
ALP (U/L)	p-nitrophenyl phosphate substrate method (JSCC improved method)
Total bilirubin (mg/dL)	Enzyme method (BOD method)
Urea nitrogen (mg/dL)	Enzyme-UV method (Urease-LEDH method)
Creatinine (mg/dL)	Enzymatic method (Creatininase-POD method)
Glucose (mg/dL)	Enzymatic method (Glck-G-6-PDH method)
Total cholesterol (mg/dL)	Enzymatic method (CO-HMMPS method)
Phospholipids (mg/dL)	Enzymatic method (COD-DAOS method)
Triglycerides (mg/dL)	Enzymatic method (GPO-HMMPS method, glycerol scavenging method)
Total protein (g/dL)	Biuret method
Protein fraction (%)	Agarose electrophoresis method
Protein fraction (g/dL)	Calculated from agarose electrophoresis and total protein
A/G ratio (%)	Agarose electrophoresis
Calcium (mg/dL)	Chlorophosphonazole III method
Inorganic phosphorus (mg/dL)	Enzyme method (PNP-XOD-POD method)
Sodium (Na, mmol/L)	Ion selective electrode method
Potassium (K, mmol/L)	Ion selective electrode method
Chlor (Cl, mmol/L)	Ion selective electrode method

Protein fractionation and A/G ratio: Epalyzer 2 (Helena Laboratories, Inc. Beaumont, TX, USA). Others: TBA-2000FR (Canon Medical Systems Inc., Tokyo, Japan).

**Table 7 nanomaterials-15-00214-t007:** Particle size distribution in the test atmosphere measured under stable conditions at the end of the exposure period.

nm	Dose (mg/m^3^)	Stage ECD (Effective Cutoff Diameter nm) Weight Ratio (%)								MMAD ^1^ (nm)	GSD ^2^	Inhalable (%)
		5800	3670	2410	1490	1090	750	340	0			
fib-CNFs	1.5	0.00	1.89	0.00	0.00	15.09	28.30	41.51	13.21	600	2.0	99.7
	7.0	1.89	3.03	4.17	2.65	17.42	24.24	34.09	12.50	800	2.4	96.9
	35	2.51	5.25	4.11	4.34	14.36	29.91	35.16	4.34	1,000	2.3	96.0

^1^ MMAD—Mass median aerodynamic diameter. ^2^ GSD—Geometric standard deviation.

**Table 8 nanomaterials-15-00214-t008:** Incidence and severity of microscopic findings (day 29).

CNF (mg/m^3^)	0	1.5	7.0	35	CNF (mg/m^3^)	0	1.5	7.0	35
Number of animals (Male)	6	6	6	6	Number of animals (Female)	6	6	6	6
Peribronchial lymph node					Peribronchial lymph node				
Aggregate, foamy macrophage					Aggregate, Foamy macrophage				
Minimal			4	5	Minimal		1	5	5
Mild					Mild				
Moderate					Moderate				
CNF (mg/m^3^)	0	1.5	7.0	35	CNF (mg/m^3^)	0	1.5	7.0	35
Number of animals (Male)	6	6	6	6	Number of animals (Female)	6	6	6	6
Lungs bronchus					Lungs bronchus				
Infiltrate, Macrophage, Alveolus					Infiltrate, Macrophage, Alveolus				
Minimal		6			Minimal		5		
Mild			6	4	Mild			6	3
Moderate				2	Moderate				3

## Data Availability

Data are contained within the article.

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
