# Peer review of "Safety of Mechanically Fibrillated Cellulose Nanofibers (CNFs) by Inhalation Exposure Based on TG412"

_nanomaterials, 2025, doi:10.3390/nano15030214_

Round 1
Reviewer 1 Report
Comments and Suggestions for Authors
Yamashita et al. present an interesting manuscript with relevant data. However, some important changes need to be made:
-First, the title of the manuscript should be revised to represent the conclusions of the study.
-The introduction should be improved so that it can have an innovative character in this field.
-In this sense, the authors should update the references used.
-The methodology is correct. Although, the authors should refer to the accepted and updated methodology and protocols.
-The authors should unify the way of presenting the results. The graphs should all be the same. This is a sign of quality and allows the uniformity necessary in a quality manuscript.
-Figure 7 should be better described. The differences between male and female are represented, although the description should be better.
-The rest of the histopathology figures should be adequately squared.
-The legend figures should be improved, and should contain the necessary information.
-The discussion should be improved with a clear translational character. Authors should justify the limitations of the study.
-At the end of the conclusions, the authors should include a graphic summary.
Comments on the Quality of English Language
The English could be improved to more clearly express the research.
Author Response
Dear reviewers
Thank you for your careful review of our original paper. We have made revisions in response to your comments, and would be grateful if you could review it again.
1)First, the title of the manuscript should be revised to represent the conclusions of the study.
The title has been changed to “Safety of mechanically fibrillated cellulose nanofibers (CNFs) by inhalation exposure based on TG412”.
2)The introduction should be improved so that it can have an innovative character in this field.
We rewrote the introduction, and focused on citing papers related to animal experiments on safety in CNFs, and described the importance of animal experiments in accordance with the OECD.
3)In this sense, the authors should update the references used.
We also re-examined the literature cited, focusing on those related to the biological safety of CNFs.
4)The methodology is correct. Although, the authors should refer to the accepted and updated methodology and protocols.
The introduction and discussion protocols have been reviewed and revised.
Until now, the safety of inhaling CNF has been tested using a single intratracheal administration method, which involves administering a large amount of CNF into the lungs at once to cause inflammation inside the lungs. This method is very simple, but it places a heavy burden on the lungs, and there are issues with the large amount of water it contains and the lack of assurance that the CNF is reaching the alveoli deep within the lungs. A more practical method is to inhale CNF as an aerosol through the nose. This is the only method that can accurately reproduce the environment in which humans are exposed to CNF over the long term. We conducted experiments in accordance with the OECD guidelines.
5)The authors should unify the way of presenting the results. The graphs should all be the same. This is a sign of quality and allows the uniformity necessary in a quality manuscript.
The appearance of Figures 5 to 7 has been standardized.
6)Figure 7 should be better described. The differences between male and female are represented, although the description should be better.
We discussed whether the difference in inflammation caused by inhalation exposure to CNFs is affected by the difference in sex between male and female rats.
7)The rest of the histopathology figures should be adequately squared.
Thank you for pointing this out. We tried to make the histopathological diagram a square, but I decided that it would be better to leave it as a 4:3 ratio diagram for balance as a figure in the paper.
8)The legend figures should be improved, and should contain the necessary information.
Graphs have been improved for easier viewing.
9)The discussion should be improved with a clear translational character. Authors should justify the limitations of the study.
The discussion has been rewritten and added to for clarity.
10)At the end of the conclusions, the authors should include a graphic summary.
A graphical summary has been added.

Reviewer 2 Report
Comments and Suggestions for Authors
This study investigated the acute toxicity of mechanically fibrillated cellulose nanofibers through a 28-day inhalation exposure experiment following OECD TG412 guidelines. No significant changes were observed in body weight, blood parameters, or biochemical profiles, the study revealed dose-dependent increases in lung weight and inflammatory responses, characterized by elevated macrophage numbers in alveoli and lymph nodes. The study determined that the No Effect Level for fib-CNFs should be below 1.5 mg/m³.
1. The collection time ranging from 5 to 300 minutes without justifying this wide variation. Define and validate the optimal collection time for each concentration.
2. The TEM should replace a high magnification. Otherwise no additional information can be seen.
3. The BALF analysis in Section 2.9 does not address the potential impact of sample processing time on cell viability.
4. Page 19, line 519: The statement about "no toxicological significance" of decreased macrophage ratio requires quantitative support. Provide specific statistical comparisons between control and exposed groups.
5. The authors fail to explain the sex-specific differences observed in neutrophil responses at 1.5 mg/m³ exposure.
6. Table 7 presents particle size distribution data without addressing the potential aggregation state of fib-CNFs.
Comments on the Quality of English Language
Good
Author Response
Dear reviewers
Thank you for your careful review of our original paper. We have made revisions in response to your comments, and would be grateful if you could review it again.
- The collection time ranging from 5 to 300 minutes without justifying this wide variation. Define and validate the optimal collection time for each concentration.
Table 3-1 and 3-2 were reversed. This has been corrected.Table 2 is a check to confirm that the expected amount of CNFs is being quantitatively supplied to the air containing solid CNFs in a nasal exposure atmosphere (1.5~35mg/m3). The balance that can be measured has a measurement accuracy of 0.01mg. If the collection time is short, the weight collected on the filter will be small, and the accuracy will decrease. On the other hand, if the collection time is too long, CNFs will accumulate on the filter and clog it, making it impossible to collect them correctly. Table 3 shows the optimal collection time for the Andersen-type filter, which collects particles of different sizes. We conducted a preliminary test to calculate the optimal collection time, and the result was the time shown in Table 3. This explanation has been added to the main text.
- The TEM should replace a high magnification. Otherwise no additional information can be seen.
As you pointed out, TEM can see the crystalline part of CNFs, but it cannot see the amorphous part, so it is not possible to obtain the correct fiber diameter. In practice, the fiber diameter was measured using FE-SEM. The photo in the figure has been changed from TEM to SEM
- The BALF analysis in Section 2.9 does not address the potential impact of sample processing time on cell viability.
The BALF test was performed immediately after the 28-day experiment was completed and the mice were dissected on the same day. The BALF test is a well-established technique in mice, rats and humans, but it does require a certain level of skill. Most of the cells collected in the BALF test are macrophages, and it is desirable to measure them within one hour of collection, so we followed this procedure.
- Page 19, line 519: The statement about "no toxicological significance" of decreased macrophage ratio requires quantitative support. Provide specific statistical comparisons between control and exposed groups.
Appendix 2 contains the raw data for BALF tests for each individual and the average. The statement that the decrease in macrophage ratio has no toxicological significance is inappropriate, and this sentence has been deleted because an absolute increase in macrophages is seen with an increase in the amount of CNFs inhaled.
- The authors fail to explain the sex-specific differences observed in neutrophil responses at 1.5 mg/m³ exposure.
The absolute number of neutrophils in BALF tests was plotted for males and females. From this, it cannot be said that there is a clear sex difference between males and females. Although it appears that there is a statistically significant difference, further verification with an increased number of individuals is necessary. This was added.
- Table 7 presents particle size distribution data without addressing the potential aggregation state of fib-CNFs.
As you pointed out, there was a lack of explanation about the actual aggregation state. This is the dried aerosol of CNFs that were exposed to the nasal cavity for one minute at 25mg/m3. It is thought that the CNFs were not inhaled as individual nanofibers, but as aggregates of spherical particles of 1μm or less, and were inhaled into the rat's lungs. There was no change in the size of the particles, only a difference in the number depending on the concentration. I have added a note about this.

Round 2
Reviewer 1 Report
Comments and Suggestions for Authors
The authors have made some changes. However, the aspects of unification of the figures have not been done. The authors should improve the histological images and the figure legends. Please correct the grammatical errors.
Comments on the Quality of English Language
The quality of English does not limit my understanding of the research.
Author Response
Dear reviewer
Thank you for your comments.
I have changed the shape of the histopathology image to a square.
I have added a legend to the figure.
I have changed the units from nanometers and micrometers to nanometers.
I have corrected the grammar and other errors in the table.
I have added a reference.
Thank you for your time and continued review.
Sincerely
Reviewer 2 Report
Comments and Suggestions for Authors
Accept
Author Response
Dear reviewer
Thank you very much for taking the time to review our paper.
Sincerely
Round 3
Reviewer 1 Report
Comments and Suggestions for Authors
Figure legends are not appropriate. Authors should show magnifications on all images and indicate techniques used on each image. Figure legends should be explanatory on all histological images.
Author Response
Dear reviewer
Thank you for your detailed advice. I have added the magnification of the photo in the figure to the description. I have also added a description of the figure. I have also standardized the position of the legend. Please check again.
Yours sincerely